# Growing Degree Day Targets for Fruit Development of Australian Mango Cultivars

Marcelo H. Amaral [1,*], Cameron McConchie [1], Geoffrey Dickinson [2] and Kerry B. Walsh [1,*]

1   Institute for Future Farming Systems, Central Queensland University, Rockhampton, QLD 4701, Australia; cmaxmail@bigpond.com
2   Department of Agriculture and Fisheries, Mareeba, QLD 4880, Australia; g.dickinson@cqu.edu.au
*   Correspondence: m.m.amaral@cqumail.com (M.H.A.); k.walsh@cqu.edu.au (K.B.W.)

**Abstract:** A forward estimate of mango (*Mangifera indica* L.) harvest timing is required for farm management (e.g., for organization of harvest labour and marketing). This forward estimate can be based on accumulated growing degree days (GDD) from an early stage of flowering to fruit harvest maturity, with fruit maturity judged on a destructive assessment of flesh colour and dry matter content. The current study was undertaken to improve GDD targets for Australian mango cultivars, to improve estimation of harvest maturity, and to document a methodology recommended for future work characterizing fruit maturation GDD for other mango cultivars. An alternate algorithm on GDD calculation involving use of a function that penalizes high temperatures as well as low temperatures was demonstrated to better predict harvest maturity in warmer climates. Across multiple locations and seasons, the required heat units (GDD, $T_b$ = 12 °C, $T_B$ = 32 °C; where $T_B$ is upper base temperature of 32 °C and $T_b$ is lower base temperature of 12 °C) to achieve maturity from asparagus stage of flowering was documented as 2185, 1728, and 1740 for the cultivars Keitt, Calypso and Honey Gold, respectively. GDD difference between the asparagus and two-thirds floral opening stages of flowering was 188 ± 18 for Calypso, 184 ± 12 for Honey Gold, 238 ± 21 for Keitt and 175 ± 10 for KP. Colour specifications for a colour card set suitable for maturity assessment of all cultivars was also proposed. A flesh colour harvest maturity card specification of 9 was proposed for the cultivar Honey Gold and 13 for the cultivar Keitt.

**Keywords:** growing degree days; flesh colour; temperature monitoring





## 1. Introduction

### 1.1. GDD

A forward estimate of mango (*Mangifera indica* L.) harvest timing is required for farm management. For example, a forward estimate of harvest date is required several months before harvest for organization of harvest resourcing, including hire of harvest labor, order of packing materials, and transport. Harvest time forecast is also essential to market planning, with longer lead time required for longer supply chains (e.g., export of Australian mango requires booking of fruit fly treatment facilities and shipping).

The first use of temperature in forecasts of mango fruit harvest maturity involved a recommendation of 1000 h above 17.9 °C (Oppenheimer, 1947; as cited in Diczbalis et al. [1]. Subsequently, cultivar specific growing degree day (GDD) requirements have been established for mango reproductive development from a given stage of flowering to harvest maturity (Table 1). The GDD calculation involves summation across days of the average of daily minimum and maximum temperature minus a 'minimum base temperature' ($T_b$), which assumes fruit development halts below this temperature, and can involve penalty for temperatures above a 'maximum base temperature ($T_B$), which assumes fruit development slows at high temperatures (see calculations in Section 1.4). A forecast of harvest

date from a given date of flowering is typically based on a historical record of daily minimum and maximum temperatures, updated with current season temperature data as the season progresses.

**Table 1.** Required heat units for fruit maturation, including upper (TB) and lower base temperature (Tb) utilized for calculation of GDD in each study, from the reproductive stages of asparagus, Christmas tree stage (2/3 flowers open, with 1/3 still not open on panicle tip) and fruit set. n/a is not applicable. The sensor location 'adjacent' refers to sensor placement adjacent to the orchard block.

| Location | Cultivar | Tb (°C) | TB (°C) | Reproductive Stage | Heat Units | Temperature Sensor Location |
|---|---|---|---|---|---|---|
| Australia [1] | Kensington Pride | 12 | n/a | asparagus | 1600 | inside canopy |
| Australia [2] | Calypso | 12 | n/a | asparagus | 1680 | on farm |
| Australia [3] | Calypso | 10 | n/a | Christmas tree | 1640 | adjacent |
| Australia [2] | Honey Gold | 12 | n/a | asparagus | 1800 | on farm |
| Australia [4] | Honey Gold | 12 | n/a | Christmas tree | 1500 | adjacent |
| Australia [2] | R2E2 | 12 | n/a | asparagus | 1800 | on farm |
| Brazil [5] | Tommy Atkins | 13 | 32 | Christmas tree | 1428 | adjacent |
| Brazil [6] | Tommy Atkins | 13 | 32 | fruit set | 1158 | adjacent |
| Brazil [7] | Alfa | 10 | n/a | Christmas tree | 2117 | 1 km from farm |
| Brazil [8] | Roxa | 10.6 | n/a | pea size fruit | 1710 | n/a |
| Brazil [9] | Uba | 10 | n/a | bud swelling | 2399 | n/a |
| Mexico [10] | Tommy Atkins | 10 | n/a | Christmas tree | 1600 | inside canopy |
| Mexico [10] | Keitt | 10 | n/a | Christmas tree | 2100 | inside canopy |
| Mexico [10] | Kent | 10 | n/a | Christmas tree | 1800 | inside canopy |
| Mexico [10] | Ataulfo | 10 | n/a | Christmas tree | 1600 | inside canopy |
| India [11] | Alphonso | 10 | n/a | fruit set | 1867 | n/a |
| India [11] | Alphonso | 17.9 | n/a | fruit set | 919 | n/a |
| India [12] | Kesar | 17.9 | n/a | fruit set | 1020 | n/a |

References: Diczbalis et al. [1], Moore [2]; Hofman et al. [3]; Winston et al. [4], Castro et al. [5], Rodrigues et al. [6], Barros et al. [7], Callejas et al. [8], Lemos et al. [9], Osuna-Garcia [10], Zagade et al. [11], Halepotara et al. [12].

GDD recommendations made to the Australian industry diverge in both the flowering stage used and the Tb value, causing some confusion in grower usage. For example, for cultivar Calypso, Moore [2] recommended 1680 GDD using a base temperature (Tb) of 12 °C from asparagus stage, while Hofman et al. [3] suggested the use of 1640 GDD on Tb of 10 °C from Christmas tree stage. Limitations to this previous work are discussed in the following sections.

### 1.2. Estimating Harvest Time from Flowering

Most mango GDD recommendations have been based on an 'eyeball' estimation that most panicles in the orchard are at a given reproductive stage, and that the fruit is ready for commercial harvest. The former estimate has a qualitative element, and the latter estimate is subject to variation on commercial and agronomic grounds. Further, existing recommendations on GDD requirements for mango maturation have generally involved work in a single season, without validation across seasons and growing conditions. As such there is a level of uncertainty in these recommendations.

For example, of the Australian work, Moore [2] based GDD recommendations on work involving tagging of trees at 'early' and 'late' flowering. Hofman et al. [3] relied on an orchard wide estimate of flowering stage in setting GDD for cultivar Calypso, with values from 1300 to 1820 units recorded at different sites and seasons. Values of the Northern Territory (NT) Australia sites only were averaged to achieve what is now an industry accepted GDD target for this cultivar (1640 units from asparagus stage on a Tb = 10 °C). Similarly, the Winston et al. [4] recommendation of 1500 units for cultivar Honey Gold fruit development was based on grower estimates of when flowering across the orchard was, on average, at Christmas tree stage.

The use of whole tree or orchard assessments of flowering and commercial harvests is convenient, but a more accurate estimate of the GDD target should be achieved by tracking of individual fruit from flowering to harvest maturity. Panicles can be tagged at a reproductive stage that has a short duration, commonly asparagus stage (Figure 1). Fruit from these panicles can be destructively harvested in the weeks before and after the date of anticipated harvest, with assessment of internal attributes used in establishing the date of optimal harvest maturity. For example, Osuna-Garcia [10] tagged individual panicles on trees with a temperature logger within the canopy in a study set in Mexico that recommended 2100–2200 GDD on a Tb of 10 °C for Keitt to reach harvest maturity from the Christmas tree stage.

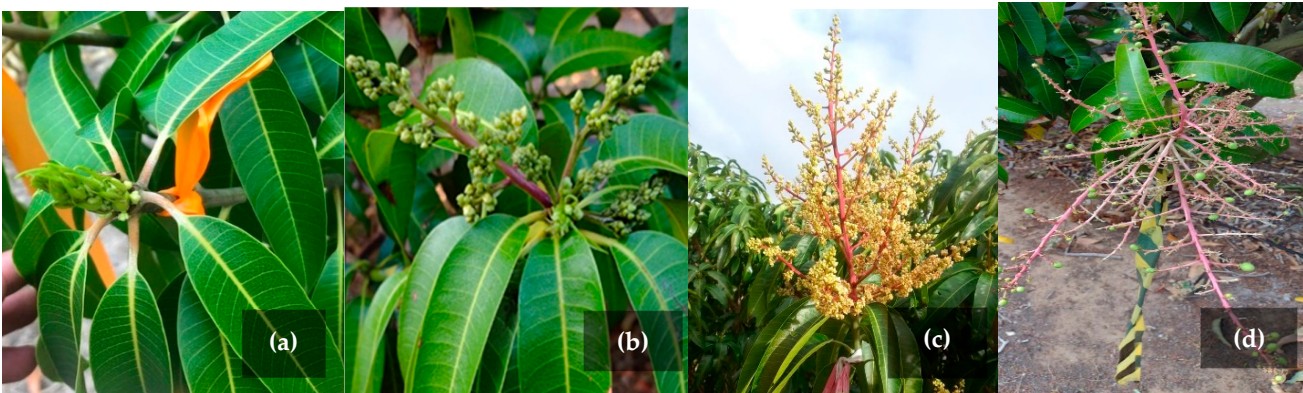

**Figure 1.** Stages of development of flowering illustrated by images of cultivar Calypso: (**a**) asparagus stage, (**b**) elongation phase, (**c**) Christmas tree stage (two thirds of flowers on panicle open), (**d**) fruit set stage.

The stages of flower development vary in duration (Figure 1). Lemos et al. [9] reported 182 to 276 GDD (equivalent to 14–21 days) from bud swelling to flower initiation (a), then 623 GDD from bud swelling to flower opening (equivalent to 49 days) (c) using Equation (1). with a Tb of 10 for mango cv. Uba. Ideally GDD estimates should be based on use of the floral stage with the shortest duration, i.e., asparagus stage (a). However, the proportion of terminals on a tree in asparagus stage is difficult to assess visually when driving through a row, and Christmas tree stage (c) is therefore usually assessed in commercial practice.

### 1.3. Estimating Harvest Maturity

The determination of when fruit is at 'harvest maturity' can be more problematic than determination of panicle development stage. 'Harvest maturity' is a commercial target which will vary by market and shelf-life needs (e.g., from a distant market served by sea-freight requiring a maximum storage potential to a local restaurant market seeking tree ripened fruit).

A range of fruit attributes change as fruit matures on tree, including skin colour, fruit shape, flesh colour, dry matter content (DMC), juice soluble solids content (SSC), titratable acidity (TA), SSC: TA and flesh firmness [13]. However, the levels of these attributes associated with a given stage of maturity can be cultivar and growing condition dependent.

Within Australian supply chains, flesh colour targets have been set for fruit harvest maturity for Kensington Pride (KP) [14] and Calypso [15]. In both cases, colour cards have been produced to assist growers in assessment of flesh colour. DMC targets have also been set for harvest of fruit in Australia (e.g., Table 2). The US National Mango Board [16] has promoted use of flesh colour as a maturity standard and has provided cultivar specific target flesh colours, and, more recently, minimum DMC values. These values are used by growers exporting to the USA and Europe from Mexico, Peru, Brazil, Costa Rica, Guatemala, and other central and South American countries (pers. comm., Agrodan, Brazil).

However, the DMC targets have been established in context of SSC and eating quality of ripened fruit, and not maturity *per se*. Fruit DMC increases with time on tree, but the absolute level of DMC can vary with growing condition, e.g., with water status [17]. Walsh et al. [18] recommend that the level of DMC associated with harvest maturity as indexed by flesh colour should be established for a given growing condition.

**Table 2.** Specifications on harvest minimum Dry Matter Content (DMC) and flesh colour for some relevant Australian grown mangoes. Flesh colours refer to colour cards produced by the named source. Kensington Pride is abbreviated to KP.

| Cultivar | DMC (%) | Source | Target Flesh Colour Card | Source |
|---|---|---|---|---|
| Calypso | 14 | Whiley and Hofman [19] | 7 | DAF [15] |
| Honey Gold | 15 | Henriod [20] | none | |
| KP | 15 | Henriod et al. [21] | single "mature" colour card | NT Farmers Association [14] |
| R2E2 | 13 | Henriod et al. [21] | none | |
| Keitt | 16 | Silva Neta [22] | 2 | National Mango Board, Orlando, FL, USA [18] |

The first Australian study to suggest GDD targets dealt with cultivar Kensington Pride. It was based on eating quality of ripened fruit, which is associated with DMC [1]. Moore [2] based GDD recommendations for all major Australian cultivars on a fruit DMC maturity specification of 14.0% (*w/w*). In the most comprehensive work undertaken on the setting of harvest maturity standards for an Australian cultivar (Calypso), Hofman et al. [3] recommended a harvest specification of a minimum DMC (14% *w/w*), flesh colour of 7 on colour score cards, SSC of 7% (*w/v*) and GDD of 1640 (from Christmas tree stage, Tb = 10 °C), with DMC, flesh colour and GDD promoted as the three most reliable attributes. Winston et al. [4] reported an attempt to use paint colour charts as references for flesh colour in Honey Gold maturity evaluation, however 'the method was discontinued in year 2 due to inconsistencies in the methods and the time involved', with preference given to the use of a GDD target established using a target DMC.

Henriod and Sole [23] established development of minimum mango harvest maturity standards for '1243', a cultivar recently released from the Australian National Mango Plant Breeding Program. Fruit were harvested at intervals around time of expected maturation on the tree and assessed for quality once ripened. It was concluded that at-harvest flesh colour (hue), SSC, TA, DMC and GDD (Tb of 12 °C, from Christmas tree stage) were all suitable maturity indicators, with minimum values for these attributes of 102 (hue), 7% *w/v*, 2.3% *w/v*, 13% *w/w* and 1040 GDD, respectively. Unfortunately, the report was not clear on the method used to record flowering (e.g., eyeball of orchard average or tagging of panicles). The study also involved a single growing location and season.

*1.4. GDD Calculation and Temperature Measurement*

Most GDD estimates for mango fruit maturation have been based on the Arnold [24] algorithm (Equation (1)), with variation in the base temperature (Tb) between 10.0 and 17.9 °C and the stage of flowering stage used (Table 1). However, two studies (Table 1) have adopted use of an upper temperature threshold (TB), as proposed by Ometto [25] (Equation (2); referred to as the 'Upper T' method in the current study). However, there is no published justification of the choice of Tb or TB values.

$$\text{GDD} = \frac{Tmax + Tmin}{2} - \text{Tb} \tag{1}$$

If TB > Tb > TM > Tm; then GDD = 0,
If TB > TM > Tm > Tb; then GDD = $\left(\frac{TMax - Tmin}{2}\right) + (Tmin - \text{Tb})$,
If TB > TM > Tb > Tm; then GDD = $\frac{(TMax - \text{Tb})^2}{2*(TMax - Tmin)}$,

$$\text{If TM > TB > Tm > Tb; then GDD} = \frac{2*(TMax-Tmin)*(Tmin-\text{Tb})+(TMax-Tmin)^2-(TMax-\text{Tb})^2}{2*(TMax-Tmin)},$$

$$\text{If TM > TB > Tb > Tm; then GDD} = \frac{1}{2}*\left[\frac{\left((TMax-\text{Tb})^2-(TMax-\text{TB})^2\right)}{TMax-Tmin}\right] \quad (2)$$

where TB is Upper base temperature, Tb is Lower base temperature, *Tmax* is maximum daily temperature and *Tmin* is minimum daily temperature.

Another limitation to previous work is the location of the temperature sensor used in calculation of GDD. Data has been used from sensors located within or outside the mango tree canopy and located adjacent to the monitored orchard or not reported assuming is from a government recording station many kilometers distant to the farm (Table 1). With branch terminals largely positioned in the outer tree canopy, use of a Bureau of Meteorology standard [26] for sensor location is recommended (i.e., placement of the temperature sensor within a white coloured weather screen with ventilated sides, positioned 1.2 m above a ground surface covered with vegetation or mulch, in an open area away from other structures by at least four times the height of those objects). Given potential variation in temperatures across a farm, placement of a sensor in the near vicinity of each monitored orchard is also recommended.

### 1.5. Cultivars

The Australian mango industry is based on the domestically developed Kensington Pride (43% of production volume), Calypso™ 'B74' (25%), R2E2 (19%) and Honey Gold™ (8%) cultivars, with minor production of Asian and Florida bred cultivars, including Keitt (information from Australian Mango Industry Association website, accessed on 1 December 2022)). Kensington, also known as KP or Bowen, was selected from a poly-embryonic line brought India to Bowen, Australia, in the late 1880s. The mono-embryonic cultivar B74 originated as a cross of Kensington Pride and of the mono-embryonic Florida variety, Sensation. Sensation is a late season cultivar that has Florida cultivars Haden and Brooks parentage. The poly-embryonic Honey Gold was selected from a Kensington Pride mother tree pollinated by an unknown cultivar in Rockhampton, QLD. The mono embryonic Keitt is a late season cultivar originating in Florida that is usually grown to extend the end of Australian mango season to late March.

### 1.6. Research Aims and Objectives

The aim of the current study is to improve existing GDD recommendations for mango reproductive development for four Australian grown cultivars (Kensington Pride, Calypso, Honey Gold and Keitt), and to provide a methodology for estimation of GDD targets for maturation of mango fruit of any cultivar, with optimization of Tb and TB values. Additionally, a comparison of a GDD calculation employing a minimum base temperature only [24] and a calculation using both a minimum (Tb) and maximum (TB) base temperature [25] is undertaken.

To improve GDD estimates over previous studies, several procedures were adopted: (i) tagging of individual panicles at an early development stage of short duration, such as asparagus stage; (ii) a time series of destructive measurements of fruit internal attributes to establish harvest timing; (iii) use of data of multiple 'calibration' sites varying in region and season; (iv) testing of recommendations at several 'validation' sites across different seasons; and (vi) use of on-farm temperature sensors positioned within six meters of an orchard block as opposed to use of more remote weather stations. In addition, given variation in colour by printers and in screen display, attention was also given to better documentation of the process of producing colour comparison cards for estimation of flesh colour.

## 2. Materials and Methods

### 2.1. Temperature Assessment

An orchard block as defined as a management unit with consistent tree cultivar, age, management history, irrigation infrastructure and harvest. Blocks typically have an area of 1 to 5 ha with more than 312 trees/ha, given average density (8 × 4) planting. For each orchard block with tagged fruit, temperature was monitored using a temperature sensor (Sensor Host, Rockhampton, Australia) in a ventilated shade screen mounted 1.2 m above covered ground, outside of the tree canopy, with temperature logged at 15 min intervals. The exceptions were the Darwin and Bungundarra site in 2018 and 2019, when the farm temperature record was used. These records were based on Hobo Onset (USA) temperature loggers within screens placed inside the tree canopy. Daily minimum and maximum temperatures were used in calculation of daily GDD.

### 2.2. Sites and Panicle Tagging Exercises

Flowering events in the 2018, 2019, and 2020 seasons were tagged, generally at asparagus stage, within 9 orchards across Australia and in Brazil (Table 3). These sites involved the cultivars dominating commercial production in Australia (KP, Calypso, R2E2 and Honey Gold), and a cultivar common to production in both Australia and Brazil (Keitt). In total, 22 populations were selected, where each population is specific in cultivar, location and date (involving 38 tagging events, given tagging of multiple flowering events in some locations) (Appendix A Table A1). The two populations of cultivar R2E2 that were tagged did not hold any fruit for two consecutive seasons. Populations from the 2018–2021 season were used in 'calibration' of GDD targets, while the 13 populations of the 2021/22 season were used in validation of proposed targets. Additional fruit from these exercises also measured non-destructively for a sizing exercise reported in Amaral and Walsh [27].

**Table 3.** Site locations.

| Region | Latitude | Longitude | Cultivars | Seasons |
|---|---|---|---|---|
| Darwin, NT | −12.754125° | 131.167722° | Calypso | 2018/19/20/21 |
| Darwin, NT | −12.548013° | 131.259296° | Honey Gold | 2021 |
| Katherine, NT | −14.615475° | 132.205328° | KP | 2020/21 |
| Katherine, NT | −14.583944° | 131.995526° | Calypso | 2020/21 |
| Katherine, NT | −14.544315° | 132.471902° | Honey Gold | 2020/21 |
| Dimbullah, QLD | −17.136831° | 145.088776° | Calypso, Honey Gold | 2020/21 |
| Bungundara, QLD | −23.025202° | 150.641147° | Honey Gold, Keitt, KP | 2018/19/20/21 |
| Belem do Sao Francisco, PE, Brazil | −8.678973° | −39.165941° | Keitt | 2020/21 |
| Curaca, BA, Brazil | −9.038435° | −39.930138° | Keitt | 2021/22 |

Panicles at several developmental stages were tagged on a single date and harvested on a single date in 2018 at Darwin site, while at the Bungundarra site, panicles at asparagus stage were recorded weekly, with all fruit harvested at a single date (Table A1). In other years, panicles were tagged at asparagus stage on a sinle date, with panicles monitored weekly for the achievement of Christmas tree stage, and resulting fruit harvested over several weeks (*n* = 20 per week) around the expected (GDD forecast) date of harvest maturation. Asparagus stage terminals were marked on the subtending vegetative stem, with the tag later moved onto the panicle when fruit set was successful for that terminal. If asparagus stage terminals were tagged on different dates (i.e., different flowering 'events'), a different colour of flagging tape was used for the different date. Asparagus stage was missed at some locations, necessitating tagging at elongation or Christmas tree stage (Table A1). At each site, typically 10 panicles were tagged on each of 10 trees, with panicles selected from around the tree canopy (Table A1). The 100 panicles typically resulted in >20 fruit, but in some cases fewer fruit were retained to harvest maturity (Table A1).

Fruit were destructively sampled at weekly intervals around the anticipated harvest maturity date anticipated from the currently recommended GDD for a given cultivar. Fruit

flesh colour (CIE LAB and hue) and oven-DM was assessed in all populations, while SSC, TA and carotenoid content (mg/kg) was assessed in some populations. These values were used to estimate the date and GDD at which harvest maturity was achieved, from the published commercial colour specifications for cultivars Calypso and Keitt. A flesh colour specification for Honey Gold was established using the existing GDD specification of 1800 units [2].

2021–2022 populations were used as validation sets for Calypso, Honey Gold and Keitt cultivars. The thirteen 2021 season populations (Table A1) were used to validate the recommended target GDD units between the stages of asparagus, Christmas tree and harvest maturity for each of the four cultivars. At each site, resulting fruit (*n* = 20) were destructively assessed for maturity attributes when the fruit reached the target GDD established in the 2018–2020 calibration exercise.

### 2.3. Estimate of Lower and Upper Base Temperature

An exercise was undertaken to optimize the Tb and TB values used in the GDD calculations. Tb from 1 to 20 °C at intervals of 1 °C were used in equation 1, and TB from 25 to 37 °C at intervals of 1 °C were used in equation 2. The Tb (°C) used while varying TB in the calculation of equation 2 was set to 12 °C. For the Tb exercise, temperature data from four different flowering events at a southern location (Bungundarra, QLD, Australia) was used. For the TB exercise, data of flowering events of northern sites (two at Darwin, NT, and one event at Katherine, NT) was used (Table 4). The sites and periods were chosen for low temperatures in assessment of Tb and high temperatures in assessment of TB.

**Table 4.** Farm locations and range of temperature values used for Tb or TB in Tb/TB optimization method. n/a is not applicable. Population # refers to numbering in Table A1.

| Region/Population # | Tb (°C) | TB (°C) | Method | Period |
|---|---|---|---|---|
| Darwin, NT/10 | 12 | 23 to 37 | Ometto, 1981 | 15/06/2021–20/10/2021 |
| Darwin, NT/11 | 12 | 23 to 37 | Ometto, 1981 | 15/06/2021–20/10/2021 |
| Katherine, NT/15 | 12 | 23 to 37 | Ometto, 1981 | 15/06/2021–20/10/2021 |
| Bugundarra, QLD/2 | 1 to 20 | n/a | Arnold, 1960 | 07/07/2020–23/12/2020 |
| Bugundarra, QLD/4 | 1 to 20 | n/a | Arnold, 1960 | 07/07/2020–23/12/2020 |
| Bugundarra, QLD/21 | 1 to 20 | n/a | Arnold, 1960 | 07/07/2020–23/12/2020 |

The method of Yang et al. [28], as adapted by Rodrigues et al. [6], was used, with the Coefficient of Variation (CV) (Equation (3)) calculated for GDD values estimated across three sites for each of Tb or TB values. The Tb or TB value with the lowest CV was chosen as the most reliable Tb or TB.

$$CV = \frac{\sigma}{\mu} * 100 \qquad (3)$$

### 2.4. Assessment of Maturity Attributes

Spectra were acquired from a mid-equatorial position on both sides of the fruit using a handheld near infrared spectrometer (F750, Felix Instruments, Camas, WA, USA). Spectra were acquired of each side of 1126 fruit (*n* = 2252 spectra). Fruit were then sliced to remove cheeks on both sides and a core of 20 mm diameter taken from the center of each slice. The skin was removed from the core, and the core then trimmed to a length of 10 mm. The flesh colour of the inside cut was assessed visually by comparison with a colour chart and by use of a Chroma Meter (CR-400, Konica Minolta, Japan) calibrated with a factory standard ceramic white tile c and set to the illumination method 'D65'. Readings were taken in the CIE LAB colour scheme. Hue angle was calculated as $\tan^{-1}$ (CIE B/CIE A) for samples with an A value above 0 and 180 $\pm$ arctan (CIE B/CIE A) for samples with A value below 0 (Ford and Roberts, 1998). Flesh colour was thus assessed at a depth of 10 mm from the skin, rather than a set distance from the stone.

One half of each core (approximately 5 g fresh weight) was used for oven-DMC analysis while the other half was diced and stored at −20 °C awaiting carotenoid analysis. For oven-DMC assessment, samples were placed on aluminum foil cups and dried in a fan forced home dehydrator EzidryFD2000(Ezidry, Adelaide, Australia) at 60 °C for 48 h [29], with weight recorded before and after using a scale of 0.001 g resolution. DMC was calculated as Dry Weight/Fresh Weight ×100. The rest of the fruit was blended and then filtered. SSC of filtrate was measured with a Bellingham and Stanley RFM320 digital refractometer and TA assessed of a 10 mL sample of juice using 0.1 N NaOH as a titrant and 1% *w/v* citric acid as a reference. TA results were expressed as citric acid equivalents.

Samples frozen at −20 °C were freeze dried (−45 °C, 200 mT) (Flexidry MP freeze drier, FTS Systems, USA) for approximately 36 h, then crushed in a ceramic mortar and pestle. Approximately 0.1 g of subsample was placed into 15 mL of acetone (99.5%, AR grade, Chem Supply, Australia), then sonicated for 30 min (Soniclean 160TD ultrasonic cleaner; Dudley Park, South Australia) and centrifuged (Heraeus Multifuge, Thermo Fisher Scientific; Sydney, Australia; $1000\times g$ for 5 min) with no volume loss reported. Total carotenoids were assessed using the method of Tomlins et al. [30], with supernatant absorbance at 450 nm measured using a UV-Vis spectrophotometer (Genesys 10S UV–Vis, Thermo Scientific, Australia). The total carotenoid concentration of the extracts (Ce, in mg/L) was calculated using the Beer-Lambert law:

$$Ce = A/\varepsilon b \times MW \times 1000 \tag{4}$$

where A is absorbance at 450 nm, $\varepsilon$ is molar absorptivity (137,400 L/mol/cm), b is path length (1 cm), MW is molecular weight of β-carotene (536.8726 g/mol). The carotenoid content of tissue (Ct, in mg/g dry weight) was calculated as:

$$Ct = Ce \times V/W \tag{5}$$

where V is volume of extract (15 mL), and W is dry weight of tissue.

### 2.5. Colour Cards

Colour cards are available to assist growers in judging flesh colour. The use of colour cards to assess sample colour can be compromised by variation in ambient lighting, issues with the users' vision (e.g., at extreme, colour blindness), and the printing process used to produce the cards and ageing of the cards. An attempt was made to quantify the colour space values of the cards, as CIE LAB value from the original pdf file as sent to printer (when available) and/or a colorimeter reading of the printed card.

Several card sets were accessed. Calypso colour cards were a product of the work of Whiley and Hofman [26]. Colour space values were accessed from the pdf files. There have been two printing runs producing card sets for grower use, and a card set from each printing run was accessed. A cultivar Kensington Pride 'business card' with harvest maturity colour was produced by NT Farmers association [14] (Darwin, Australia). Keitt colour swatches as electronically published by the US National Mango Board [18] were also accessed.

### 2.6. Statistical Analysis

Linear correlations between parameters and one-way ANOVA statistical analysis were undertaken using the Rstudio 4.1.2 (Boston, MA, USA). A significance *p*-value < 0.05 was adopted. Population results were expressed as mean ± SD., or mean ± SE for all parameters (DMC, TSS, CIE-Lab, hue, TA and SSC: TA ratio).

## 3. Results and Discussion

### 3.1. GDD Algorithm-Choice of Tb and TB

The use of a different Tb merely creates a daily offset in the GDD increment when the daily average of *Tmax* and *Tmin* exceeds Tb. For example, if Tb is decreased from

12 to 10 °C, then 2 extra units will be accumulated every day. In this scenario, the use of different Tb values requires a revised value for the heat units associated with harvest maturity, but the date the GDD target is achieved is not affected. However, if the daily average of *Tmax* and *Tmin* is less than Tb, differences in the estimate of the date of harvest maturity emerge.

The CV on calculated GDD across three populations from a cooler growing area was minimal at a Tb of 12 or 13 °C (Figure 2). A Tb of 12 °C is therefore recommended based on low CV and its current common use as a base temperature in most Australian GDD calculations. For TB, minimal CV occurred at 32 °C (Figure 3). In comparison, Rodrigues et al. [6] recommended a *Tmin* of 13 °C and a TB of 32 °C for mango cv. Tommy Atkins in Brazil.

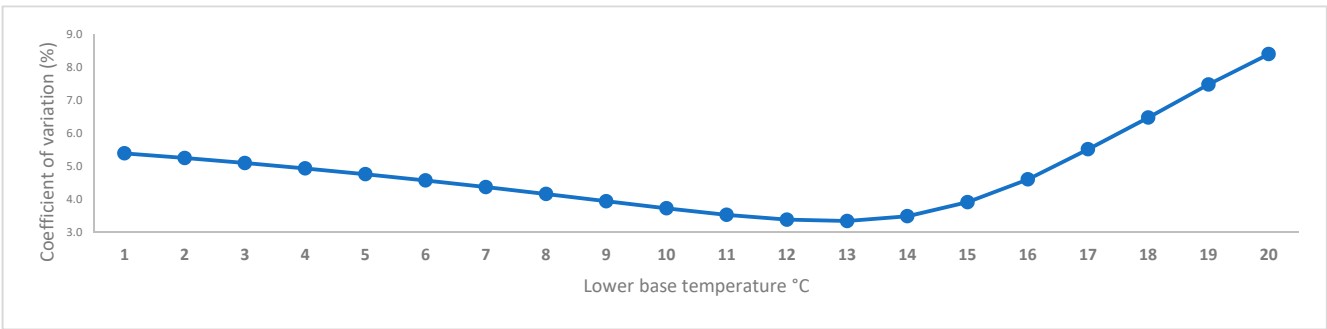

**Figure 2.** Coefficient of variation for GDD calculated using different lower base temperatures (Tb). Data of three flowering events of the southern-most farm of this study, Bungundarra, QLD (Table A1).

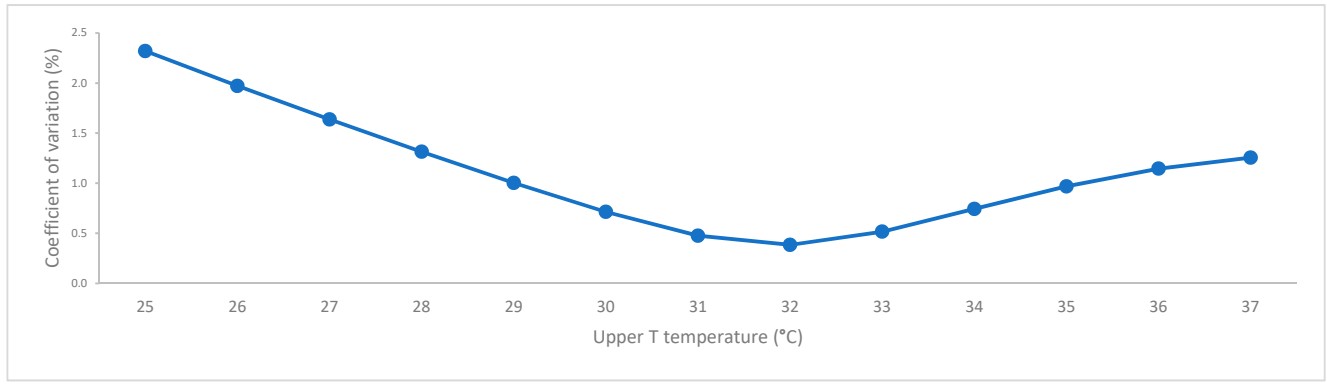

**Figure 3.** Coefficient of variation for GDD calculated using different upper base temperatures (TB) and a Tb of 12°C. Data of three flowering events of the northern-most farms of this study, in Darwin and Katherine, NT (Table A1).

### 3.2. GDD Algorithm Implementing TB

The use of a TB in a GDD calculation (Equation (2)) provided target harvest dates up to 13 days later than the standard GDD calculation (Equation (1)) for NT sites, up to 5 days later in Far North Queensland sites and up to 3 days later in Central Queensland sites (Table 5). This difference between sites mirrors differences in the proportion of days with temperatures above 32 °C.

**Table 5.** Data showing 2021/22 harvest date estimated from flowering date using the standard and the Upper T methods of calculating GDD for 9 populations, varying in location and cultivar. GDD (from asparagus stage, Tb = 12 °C, TB = 32 °C) targets of 1680 for Calypso, 1800 for Honey Gold (HG) and 1600 for Kensington Pride (KP) were used.

| Region-Cultivar-Population # | Flowering Date | Harvest Date (Standard GDD) | Harvest Date (Upper T GDD) | Difference in Days |
|---|---|---|---|---|
| Darwin NT-Calypso-10 | 4-Jun | 18-Sep | 1-Oct | 13 |
| Darwin NT-HG-12 | 8-Jul | 25-Oct | 4-Nov | 10 |
| Katherine NT-Calypso-14 | 15-Jun | 14-Oct | 22-Oct | 8 |
| Katherine NT-KP-15 | 15-Jun | 2-Oct | 9-Oct | 7 |
| Katherine NT-HG-13 | 2-Jul | 23-Oct | 4-Nov | 12 |
| Dimbulah FNQ-Calypso-16 | 30-Jun | 22-Nov | 26-Nov | 4 |
| Dimbulah FNQ-HG-17 | 30-Jun | 29-Nov | 4-Dec | 5 |
| Bungundarra CQ-KP-20 | 5-Jun | 26-Nov | 29-Nov | 3 |
| Bungundarra CQ-HG-21 | 5-Jun | 16-Dec | 18-Dec | 2 |

\# refers to Table A1 population numbers.

While the use of the Upper T method impacts the predicted harvest date, it remains to be demonstrated that the physiological premise is correct (i.e., that maturation slows at temperatures above 32 °C). This demonstration is attempted in the following section.

### 3.3. GDD between Reproductive Stages

A sample of 15 panicles tagged at asparagus stage of a range of Honey Gold, Calypso, Keitt, and Kensington Pride populations were observed either weekly or every three days and the date that Christmas tree stage was reached was recorded (Table 6). The mean and standard deviation, across locations and seasons, on the GDD difference between the two flowering stages was $188 \pm 18$ for Calypso, $184 \pm 12$ for Honey Gold, $238 \pm 21$ for Keitt and $175 \pm 10$ for KP (Table 6). These estimates should be more accurate than the recommendation of 300 units for all cultivars, as given by Moore [2], given that the latter estimate was based on grower estimates of the date of 'early' and 'late' flowering stages, rather than from tracking of individual panicles. The difference between the GDD requirement of the three Australian cultivars, which share parentage, and the Florida cultivar, Keitt, is consistent with a genetic component to the GDD requirement.

**Table 6.** GDD (Tb 12 °C, TB 32 °C) between asparagus and Christmas tree stages ($n$ = 15), with average and SD for each cultivar. Population numbers refer to Table A1.

| Population # | Cultivar | Date of Christmas Tree Stage | GDD |
|---|---|---|---|
| 2018 | | | |
| 1 | Calypso | 13-Jun | 164 |
| 2020 | | | |
| 5A | Honey Gold | 5-Aug | 180 |
| 5B | Honey Gold | 29-Aug | 171 |
| 6A | Honey Gold | 5-Aug | 180 |
| 6B | Honey Gold | 9-Sep | 183 |
| 7 | Keitt | 1-Sep | 237 |
| 8 | Keitt | 16-Sep | 231 |
| 9 | Keitt | 29-Aug | 297 |
| 2021 | | | |
| 10 | Calypso | 18-Jun | 180 |
| 12 | Honey Gold | 24-Jun | 186 |
| 13 | Honey Gold | 17-Jul | 176 |
| 14 | Calypso | 2-Jul | 193 |
| 15 | KP | 30-Jun | 184 |
| 16 | Calypso | 23-Jul | 214 |

**Table 6.** *Cont.*

| Population # | Cultivar | Date of Christmas Tree Stage | GDD |
|:---:|:---:|:---:|:---:|
| 17 | Honey Gold | 23-Jul | 214 |
| 20 | KP | 28-Jul | 165 |
| 21 | Honey Gold | 21-Jul | 184 |
| 22 | Keitt | 4-Aug | 235 |
| | | Average ± SD | |
| | Calypso | | 188 ± 18 |
| | Honey Gold | | 184 ± 12 |
| | Keitt | | 238 ± 21 |
| | KP | | 175 ± 10 |

*3.4. Colour Cards for Flesh Colour Assessment*

A strong correlation between fruit GDD and CIE B existed across cv. Calypso (R = 0.96, *n* = 513), cv. HoneyGold (R = 0.90, *n* = 611) and cv. Keitt (R = 0.94, *n* = 240) populations. In addition, CIE B was strongly correlated to organoleptic parameters such as SSC:TA ratio ($R^2$ = 0.89), TA ($R^2$ = 0.84), DMC (%$w/w$) ($R^2$ = 0.86), and total carotenoids content ($R^2$ = 0.86), although poorly correlated to SSC (%$w/w$) ($R^2$ = 0.58) (Appendix B). Therefore, flesh colour is recommended as the primary index in assessment of harvest maturity.

The flesh colour of cut fruit was matched by visual comparison to colour cards. Calypso and KP fruit judged as matching Calypso colour card 7 (CIE B = 32) had a mean CIE B value of 32.9 (with range 30.2 to 35.6) and 32.2 (range 29.4 to 35.0), respectively. Keitt fruit judged as matching Keitt colour card 2 (B = 51) had a mean CIE B value of 51.0 (range 45.9 to 55.0) (Table A2). Human sorting was thus successful in matching fruit to colour cards.

CIE LAB values varied between the pdf associated values and readings taken of cards from different print runs using different printers, although values for a given maturity value were consistent for different prints from the one printer (Table A2).

A set of swatches with colour values spanning the range associated with harvest maturity of all cultivars involved in this study was proposed, with indication of the swatch associated with maturity of each cultivar (Table A3). CIE LAB values are given as expected readings of fruit flesh using a calibrated colorimeter such as the Minolta CR400 (from Tables 5–7). A second CIE L value is given in Table A3, being the value in a pdf electronic file to achieve desired CIE LAB values in a print made by an office printer (Bizhub C4000i, Konica Minolta, Japan). These values were determined by trial and error. A similar optimization is recommended when using other printers.

*3.5. Cultivar Specifications on Maturity*

3.5.1. Time Course of Maturity Attributes

Fruit attributes were destructively assessed for fruit from fruit stone-hardening stage to past commercial harvest for several populations (Table A1). One example each of Calypso, Honey Gold and Keitt populations is given graphically in Figure 4 (other data is presented in Amaral [31]. In all cultivars, SSC showed little change with time on trees (Figure 4). DMC, TA, and flesh colour, as indexed by CIE B value or hue, changed as fruit matured, while CIE A value changed only in Keitt fruit (Figure 4). Change was not linear, with perturbations likely due to changes in growing conditions, particularly water status (e.g., Anderson et al. [16]).

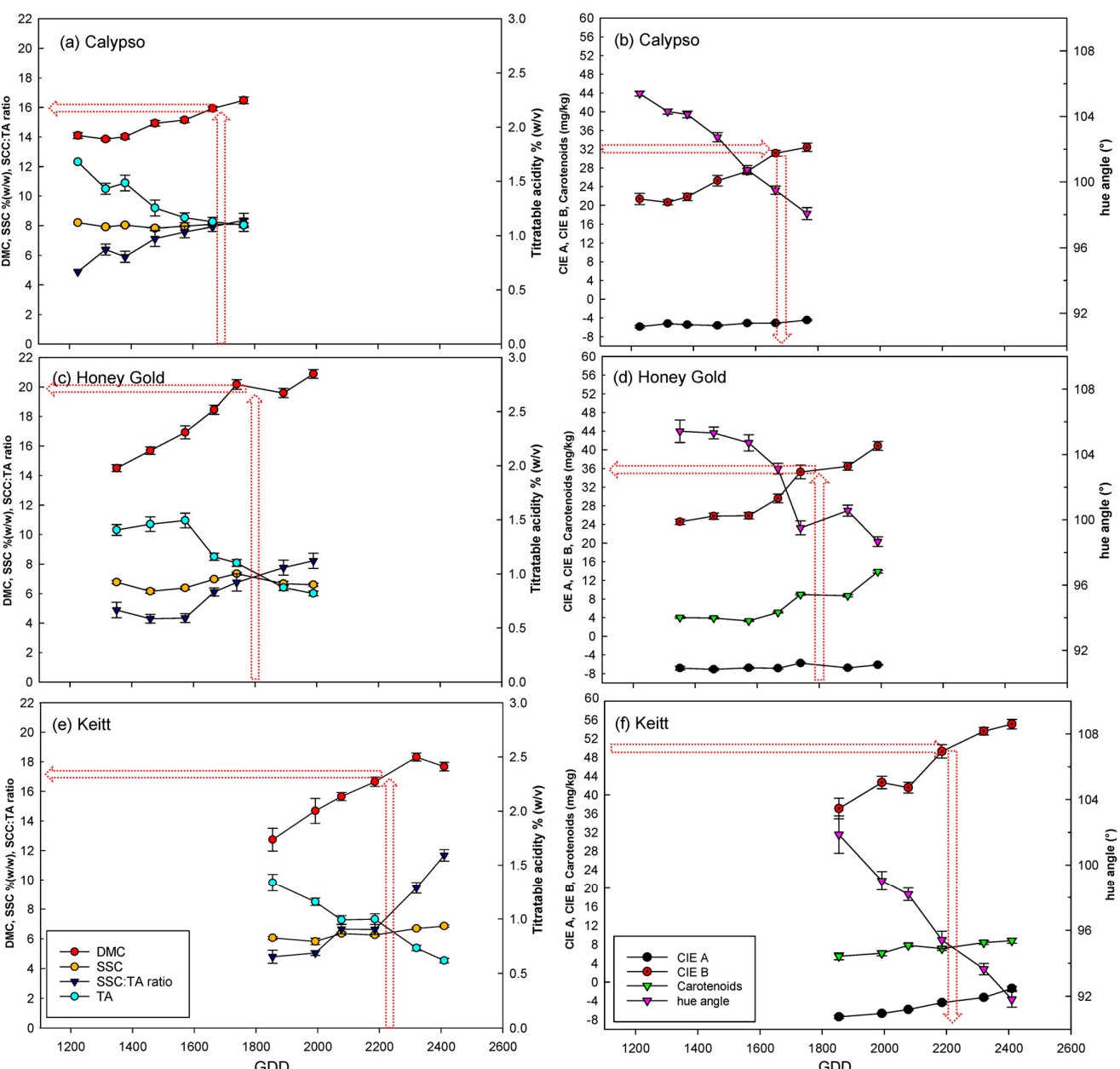

**Figure 4.** Time course from stone hardening of fruit attributes (for populations 3, 5 and 7 of Table A1, shown in top (**a**,**b**), middle (**c**,**d**) and bottom (**e**,**f**) panels, respectively). Left panels: DMC, SSC, SSC:TA ratio and TA. Right panels: flesh colour CIE AB values and total carotenoids (mg/kg); Data presented as mean with associated standard error (*n* = 10 to 20 fruit for Calypso, 5 to 10 fruit for Honey Gold and Keitt). For Calypso and Kensington Pride, red lines are drawn from the CIE B value associated with harvest maturity, i.e., 32 for Calypso and Kensington Pride and 51 for Keitt, to the associated GDD (lright panel), and from this GDD value to the associated DMC value (left panel). For Honey Gold, the red line is drawn from the recommended GDD of 1800 to the associated B and DMC values. *X* axis is given in heat units (GDD) estimated from asparagus stage of flowering using Tb 12 °C and TB 32 °C.

In the example Calypso population, CIE B value increased at about 2.3 units per week while DMC increased at approximately 0.52% *w/w* per week (Figure 4). The B value harvest maturity target (32 units) was reached at 1664 GDD from asparagus stage and a DMC of 16.0% *w/w* (see red arrows on Figure 4). For a Keitt population, B value increased at about 4 units per week and DMC increased at approximately 1.0% *w/w* per week. The B value

target of 51 units was reached at 2210 GDD from asparagus stage with a fruit DMC of 17.0% *w/w*, as estimated by linear interpolation of adjacent values.

The commercially accepted maturity target for Honey Gold is a GDD of 1500 from the Christmas tree stage [4] or 1800 from the asparagus stage [2]. For a Honey Gold population, B value increased at about 2.7 units per week and DMC increased at approximately 1.40% *w/w* per week (Figure 4). The harvest maturity GDD target of 1800 was achieved when fruit flesh B value was 36 and fruit DMC was 20.0% *w/w* (see red arrows on Figure 4).

This procedure was followed with all calibration data sets, to associate attribute levels to the available harvest maturity targets of flesh colour in Calypso, KP and Keitt, and to the GDD target for Honey Gold, as described in the following sections. While the focus has been given to GDD and CIE B, levels of DMC, TA, and SSC:TA ratio are also reported in the following sections.

### 3.5.2. Calypso

For a 2018 Calypso population, CIE B = 32 was reached at 1720 GDD and a DMC of 16.5% *w/w* (Table 7), based on interpolation between measurements made at 1692 and 1792 GDD. For the same orchard in 2019, CIE B = 32 was reached at GDD 1735 with a DMC of 16.3% *w/w*. From the average of data of these two populations, a GDD of 1728 from asparagus stage is recommended to achieve a flesh B value of 32, with an associated DMC of 16.4% *w/w*, TA of 0.86% *w/w* and SSC:TA ratio of 10.6. This represents an increase of approximately 3 (summer) calendar days over the currently recommended GDD of 1640 on Tb 10 °C [3] or 1680 on Tb 12 °C [2].

**Table 7.** Observed values of Calypso fruit attributes at the two harvest dates with CIE B value bracketing the target value of 32 (in italics), and values at the target value, in italics, as estimated by interpolation. GDD are calculated from asparagus stage with Tb = 12 °C, and TB = 32 °C for fruit from the same orchard in 2018 and 2019 seasons. Panicles were tagged on different dates and harvested on one date. Population numbers refer to Table 4.

|  | CIE B | Hue | DMC | TA | SSC:TA Ratio |
|---|---|---|---|---|---|
| **2018 (pop # 1)** |  |  |  |  |  |
| GDD 1692 | 28.0 ± 0.6 | 99.6 ± 0.2 | 15.4 ± 0.2 | 0.90 ± 0.03 | 8.4 ± 0.2 |
| *GDD 1720* | *32* | *96.5* | *16.5* | *0.8* | *13* |
| GDD 1792 | 38.3 ± 1.1 | 93.6 ± 0.6 | 18.0 ± 0.3 | 0.40 ± 0.10 | 20.0 ± 1.4 |
| **2019 (pop # 3)** |  |  |  |  |  |
| GDD 1664 | 31.1 ± 0.6 | 99.5 ± 0.3 | 15.9 ± 0.1 | 1.12 ± 0.10 | 7.9 ± 0.3 |
| *GDD 1735* | *32* | *98.7* | *16.3* | *1.1* | *8.2* |
| GDD 1765 | 32.4 ± 0.9 | 98.1 ± 0.4 | 16.5 ± 0.2 | 1.10 ± 0.10 | 8.3 ± 0.5 |

# Refers to population numbers in Table A1.

### 3.5.3. Keitt

In Keitt fruit population #7 (Table 6), the target maturity CIE B value of 51 was achieved at a DMC of 17.0% *w/w*, a TA of 0.85% *w/w*, an SSC:TA ratio of 7 and a total carotenoids content of 7.5 mg/kg at 2210 GDD from asparagus stage (Table 6). From the data of three populations (Table 8), a GDD of 2185 was chosen as a minimum value recommended to achieve a minimum flesh CIE B value of 51.0.

**Table 8.** Observed values of Keitt fruit attributes at two or three harvest dates with CIE B value bracketing the target value of 51, and values at the target value, in italics, as estimated by interpolation. GDD are calculated from asparagus stage with Tb = 12 °C, and TB = 32 °C for fruit of three populations. Panicles were tagged on one date and harvested on a range of dates. Population numbers refer to Table 4.

| | CIE B | Hue | DMC (%*w/w*) | TA (%*w/v*) | SSC:TA Ratio | Carotenoid (mg/kg) | Flesh Card Colour |
|---|---|---|---|---|---|---|---|
| Pop # 7 | | | | | | | |
| GDD 2185 | 49.3 ± 1.5 | 95.4 ± 0.5 | 16.7 ± 0.3 | 1.00 ± 0.04 | 6.6 ± 0.3 | 7.1 ± 0.4 | 1.9 ± 0.1 |
| *GDD 2210* | *51* | *94.8* | *17* | *0.85* | *7* | *7.5* | *2* |
| GDD 2320 | 53.6 ± 0.8 | 93.6 ± 0.3 | 18.3 ± 0.3 | 0.73 ± 0.02 | 9.5 ± 0.4 | 8.4 ± 0.4 | 2.3 ± 0.1 |
| GDD 2410 | 55.1 ± 1.0 | 91.8 ± 0.5 | 17.7 ± 0.3 | 0.62 ± 0.01 | 11.6 ± 0.4 | 8.8 ± 0.5 | 2.3 ± 0.1 |
| Pop # 8 | | | | | | | |
| GDD 2007 | 43.2 ± 1.9 | 98.5 ± 0.6 | 16.0 ± 0.6 | 1.12 ± 0.13 | 6.0 ± 0.5 | 4.7 ± 0.6 | 1.9 ± 0.2 |
| GDD 2142 | 51.0 ± 1.2 | 94.4 ± 0.6 | 16.7 ± 0.2 | 1.05 ± 0.04 | 6.6 ± 0.3 | 8.4 ± 0.6 | 2.4 ± 0.2 |
| GDD 2233 | 51.1 ± 2.0 | 92.7 ± 0.7 | 16.4 ± 0.4 | 1.05 ± 0.04 | 7.5 ± 0.2 | 7.5 ± 0.6 | 2.7 ± 0.2 |
| Pop # 9 | | | | | | | |
| GDD 2297 | 55.0 ± 1 | 95.1 ± 0.4 | 15.9 ± 0.1 | 0.77 ± 0.02 | 10.6 ± 0.4 | 10.2 ± 0.5 | 2.1 ± 0.1 |
| GDD 2350 | 54.5 ± 1.1 | 95.1 ± 0.4 | 16.2 ± 0.2 | 0.81 ± 0.02 | 10.8 ± 0.3 | 9.7 ± 0.6 | 2.1 ± 0.1 |
| GDD 2452 | 57.0 ± 0.7 | 94.8 ± 0.4 | 16.5 ± 0.2 | 0.70 ± 0.03 | 11.7 ± 0.5 | 10.7 ± 0.7 | 2.6 ± 0.1 |

\# Refers to population numbers in Table A1.

### 3.5.4. Honey Gold

For Honey Gold, a GDD of 1800 from asparagus stage (on Tb = 12 °C) is a recommended maturity specification [2]. At 1800 GDD, fruit of population 2 reached a CIE B of 36 (interpolation using 4 data points) and a DMC of 17.6% *w/w* (Table 9). For the same orchard in the next season, fruit at 1800 GDD possessed a CIE B of 40.5 (extrapolated from 3 data points) and a DMC of 24.4% *w/w*. The difference in values was associated with unusual growing conditions, as the orchard was in a declared bushfire disaster area in November and was subject to high temperatures and dry conditions. In 2020, tagging occurred on two flowering events (FE) on each of two orchards on the one farm (populations 5 and 6). In FE1 of orchard 1, a GDD of 1800 was associated with a CIE B of 34 and a DMC of 18.5% *w/w*, while in FE2, it was associated with a CIE B of 35.5 and a DMC of 19.5% *w/w*. In FE1 of orchard 2, a GDD of 1800 was associated with a CIE B of 34 and a DMC of 18.5% *w/w*, while in FE2, it was associated with a CIE B of 34 and DMC of 19.2% *w/w*. Averaged across these populations, 1800 GDD was associated with CIE B of 36 ± 2.1 (mean ± SD) and DMC of 19.5 ± 2.2. The CIE B value of 36 is therefore recommended as a flesh colour standard for Honey Gold. This value is equivalent to the DAF 2019 print of Calypso colour card 9.

The preceding discussion is based on the Moore (2010) recommendation of a GDD of 1800 from asparagus stage or a GDD of 1500 from Christmas tree stage (on Tb = 12 °C) as a maturity specification for Honey Gold. Winston et al. [4] confirmed a GDD (on Tb = 12 °C) of 1500 from the Christmas tree stage. However, the 300 GDD asparagus to Christmas stage difference used by Moore [2] was an approximation used across all cultivars. A GDD of 184 was established in the current study for development between the two flowering stages. An average GDD of 1560 was estimated from Christmas tree stage to harvest maturity, suggesting a GDD of 1560 + 184 = 1744 should be used for forecasting of Honey Gold harvest maturity from asparagus stage.

**Table 9.** Observed values of Honey Gold fruit attributes at several harvest dates bracketing the GDD (Tb = 12 °C) target of 1800 units from asparagus stage, and values at the target value, in italics, as estimated by interpolation or extrapolation. Panicles were tagged at asparagus stage on different dates. Population numbers refer to Table 4. 'n/a' is not available.

| | CIE B | Hue | DMC (% *w/w*) | TA (% *w/w*) | SSC:TA Ratio | Carotenoid (mg/kg) |
|---|---|---|---|---|---|---|
| 2018 CQ pop 2 | | | | | | |
| GDD 1658 | 34.1 ± 0.8 | 101.8 ± 0.3 | 16.3 ± 0.2 | 1.00 ± 0.02 | 5.7 ± 0.1 | n/a |
| GDD 1791 | 38.2 ± 3.1 | 99.4 ± 0.9 | 18.0 ± 0.4 | 0.87 ± 0.08 | 7.4 ± 0.7 | n/a |
| *GDD 1800* | *36* | *100.3* | *17.6* | *0.95* | *6.8* | n/a |
| GDD 1816 | 34.7 ± 0.7 | 100.6 ± 0.2 | 17.3 ± 0.2 | 0.95 ± 0.02 | 6.3 ± 0.1 | n/a |
| GDD 1850 | 38.7 ± 0.7 | 99.8 ± 0.2 | 18.1 ± 0.1 | 0.85 ± 0.02 | 7.3 ± 0.2 | n/a |
| 2019 CQ pop 4 | | | | | | n/a |
| GDD 1638 | 34.8 ± 0.7 | 96.8 ± 0.4 | 22.6 ± 0.3 | 1.49 ± 0.04 | 6.3 ± 0.2 | n/a |
| GDD 1691 | 35.7 ± 0.7 | 96.6 ± 0.4 | 22.7 ± 0.2 | 1.45 ± 0.05 | 6.2 ± 0.1 | n/a |
| GDD 1756 | 39.2 ± 0.6 | 96.2 ± 0.2 | 24.1 ± 0.2 | 1.33 ± 0.03 | 6.8 ± 0.1 | n/a |
| *GDD 1800* | *40.5* | *96* | *24.4* | *1.2* | *6.8* | n/a |
| 2020 CQ pop 5 Orchard 1 | | | | | | |
| * GDD 1747 | 32 ± 1.3 | 101 ± 0.3 | 17.2 ± 0.3 | 1.15 ± 0.06 | 6.2 ± 0.3 | 5.0 ± 0.5 |
| * *GDD 1800* | *34* | *101* | *18.5* | *1.15* | *6.2* | *6.1* |
| * GDD 1850 | 35.6 ± 1.2 | 100.5 ± 0.4 | 19.4 ± 0.4 | 1.15 ± 0.02 | 6.3 ± 0.2 | 6.9 ± 0.5 |
| ** GDD 1740 | 35.3 ± 1.5 | 99.5 ± 0.4 | 20.2 ± 0.3 | 1.1 ± 0.02 | 6.8 ± 0.2 | 9.0 ± 0.6 |
| ** *GDD 1800* | *35.5* | *100.5* | *19.5* | *1* | *7* | *8.5* |
| ** GDD 1892 | 36.5 ± 0.9 | 100.6 ± 0.3 | 19.6 ± 0.3 | 0.87 ± 0.02 | 7.8 ± 0.2 | 8.7 ± 0.5 |
| 2020 CQ pop 6 Orchard 2 | | | | | | |
| * GDD 1747 | 32.9 ± 0.6 | 103 ± 0.3 | 17.1 ± 0.2 | 1.17 ± 0.03 | 5.9 ± 0.3 | 3.3 ± 0.3 |
| * *GDD 1800* | *36.5* | *101* | *18.2* | *1.13* | *6.3* | *6.5* |
| * GDD 1850 | 39.9 ± 1.1 | 99.8 ± 0.3 | 19.1 ± 0.2 | 1.09 ± 0.02 | 6.6 ± 0.1 | 5.1 ± 0.3 |
| ** GDD 1687 | 31.4 ± 1.1 | 101.2 ± 0.3 | 19.1 ± 0.3 | 1.15 ± 0.04 | 6.0 ± 0.2 | 7.7 ± 0.5 |
| ** *GDD 1800* | *34* | *100.5* | *19.2* | *1.18* | *5.6* | *8* |
| ** GDD 1839 | 34.3 ± 1 | 100.3 ± 0.3 | 19.2 ± 0.5 | 1.19 ± 0.04 | 5.5 ± 0.2 | 8.5 ± 0.6 |

* Represents the first flower event one for the given orchard; ** Represents the second flower event for the given orchard.

### 3.6. GDD Validation

The GDD targets established in the preceding section for Calypso and Keitt (1728 and 2185, respectively), associated with flesh CIE B values of 32 and 51, respectively, were trialed in 2021 validation exercises. The Calypso target was also used for KP. Across multiple populations harvested at these GDD values, flesh CIE B value met the colour specification of 32 for all six Calypso populations, and the specification of 51 in five of the six Keitt populations (Table 10). DMC met the specification minimum of 14.0% *w/w* in all cases, varying from 15.4 to 16.7% *w/w* across the six Calypso populations, and 14.0 to 17.4% *w/w* across the six Keitt populations.

**Table 10.** GDD validation exercise: Maturity attributes of fruit 2021/22 season harvested close to a GDD of 1728, 1740, 2185, and 1600 for Calypso, Honey Gold, Keitt and KP populations, respectively. Population numbers refer to Table A1.

| Pop # | Season | Fruit | CIE B | Hue | DMC (%) | Colour Cards | GDD |
|---|---|---|---|---|---|---|---|
| Calypso | | | | | | | |
| 1 * | 2018 | 95 | 32.0 | 96.5 | 16.5 | 7.0 | 1720 |
| 3 * | 2019 | 209 | 32.0 | 98.7 | 16.3 | 7.0 | 1735 |
| 10 | 2021 | 88 | 31.0 | 100.0 | 16.0 | 6.0 | 1757 |
| 11 | 2021 | 54 | 30.0 | 101.0 | 16.0 | 6.6 | 1700 |
| 14 | 2021 | 18 | 31.0 | 97.9 | 16.7 | 6.1 | 1741 |
| 16 | 2021 | 48 | 31.0 | 100.0 | 15.4 | 6.0 | 1757 |
| Mean ± SE | | | 31.2 ± 0.3 | 99 ± 0.6 | 16.2 ± 0.2 | 6.2 ± 0.1 | 1735 ± 10.1 |
| Honey Gold | | | | | | | |
| 2 * | 2018 | 186 | 35.0 | 100.0 | 17.5 | 8.0 | 1740 |
| 4 * | 2019 | 96 | 38.3 | 96.5 | 22.9 | 9.0 | 1740 |
| 5a | 2020 | 110 | 32.0 | 101.0 | 17.2 | 7.2 | 1747 |
| 5b | 2020 | 114 | 35.3 | 99.5 | 20.2 | 8.5 | 1740 |
| 6a | 2020 | 116 | 32.9 | 103.0 | 17.1 | 7.3 | 1747 |
| 6b | 2020 | 94 | 33.5 | 101.0 | 18.8 | 8.0 | 1740 |
| 12 | 2021 | 26 | 34.0 | 98.7 | 18.9 | 8.0 | 1732 |
| 13 | 2021 | 80 | 37.0 | 98.5 | 17.2 | 8.7 | 1733 |
| 17 | 2021 | 30 | 34.9 | 100.0 | 16.2 | 8.0 | 1757 |
| 21 | 2021 | 30 | 29.0 | 103.2 | 17.9 | 6.1 | 1740 |
| Mean ± SE | | | 34.2 ± 0.8 | 100.1 ± 0.6 | 18.4 ± 0.6 | 7.7 ± 0.2 | 1741.6 ± 2.2 |
| Keitt | | | | | | | |
| 7 | 2020 | 142 | 51.0 | 95.4 | 17.0 | 2.0 | 2230 |
| 8 | 2020 | 98 | 51.0 | 94.4 | 16.8 | 2.0 | 2142, 2233 |
| 9 | 2020 | 162 | 54.9 | 96.1 | 15.9 | 2.1 | 2297 |
| 18 | 2021 | 30 | 55.0 | 101.9 | 14.0 | 1.9 | 2185 |
| 19 | 2021 | 30 | 55.1 | 96.8 | 15.3 | 2.1 | 2188 |
| 22 | 2021 | 10 | 48.4 | 96.2 | 17.4 | 2.0 | 2185 |
| Mean ± SE | | | 52.5 ± 1.2 | 96.5 ± 0.8 | 16.1 ± 0.3 | 2.0 ± 0.0 | 2209 ± 8.0 |
| KP | | | | | | | |
| 15 | 2021 | 26 | 32.0 | 99.4 | 18.2 | 6.7 | 1602 |
| 20 | 2021 | 4 | 26.0 | 104.8 | 14.7 | 5.0 | 1638 |
| Mean ± SE | | | 29.0 ± 1 | 102.1 ± 1.9 | 16.5 ± 1.2 | 6.0 ± 0.6 | 1620 ± 12.7 |

* Populations marked with an asterisk were not used in the validation set as results were based on interpolated data for Calypso based on CIE B 32 and for Honey Gold on GDD of 1740. # Refers to population codes in Table A1.

Of the two KP populations, one failed to meet the B value specification of 32 (Table 8). DMC met the specification minimum of 14.0% *w/w* in both cases, at 14.7 and 18.2% *w/w*. The failure of one KP population to achieve CIE B specification at GDD 1600 is attributed to a sampling issue. Only two fruits remained from an initial tagging of 100 panicles for each event, and those fruits were located inside the canopy. Further work is required to confirm the recommended KP GDD.

The recommended Calypso GDD target (on Tb 12 °C, from asparagus) of 1728 is 48 units greater than the Moore [2] specification of 1680. This difference will be achieved in four days in the hotter temperatures prevailing near harvest.

The recommended Keitt GDD target (on Tb 12 °C, from asparagus) of 2185 is much greater than the Moore [2] specification of 1680, but it is consistent with the recommendation of Osuna-Garcia [10] of between 2100 and 2200 GDD (average 2150) on a Tb of 10 °C for Keitt to reach harvest maturity from Christmas tree stage. At an average GDD accumulation rate of 18 units per day, 2150 units is achieved in 107 days. Assuming *Tmin* is always >12 °C, the equivalent GDD on a Tb = 12 °C is 2150 − (107 × 2) = 1936. Adjusting for the asparagus to Christmas tree development time yields a GDD requirement of

1936 + 240 = 2176 on Tb of 12 °C from asparagus stage. This value is consistent with the recommendation of the current study (2185).

Honey Gold fruit harvest targeted a GDD of 1744. In practice, harvests of the four validation populations occurred at GDD values ranging from 1732 to 1757, with an average of 1740 (Table 8). The flesh colour target of CIE B = 36 for Honey Gold, as established in the calibration exercise, was validated on the 2021 data. B values between 34 and 37 were achieved across four 2021 validation populations, however one population (pop 21) achieved a value of only 29. This result is attributed to the low number of samples in this population ($n$ = 5) which resulted from a chilling event injuring 95% of the tagged panicles. The few remaining panicles were positioned inside the canopy and can be expected to experience a cooler microclimate and delayed maturation. Fruit DMC met the specification minimum of 14% $w/w$ in all populations, varying from 16.2 to 22.9% $w/w$ across the eight populations. The GDD target of 1744 is therefore recommended for use with Honey Gold, being associated with a B value of 36 (ranging from 34 to 37). This is a relatively small change on the previously recommended GDD of 1800, being equivalent to a three (summer) calendar days earlier harvest.

## 4. Conclusions

A recommendation of a methodology to follow in establishing the GDD of fruit development of mango cultivars has been presented. The Upper T temperature method (Equation (2)), using a Tb = 12 °C and TB = 32 °C, is recommended over the standard method for estimation of GDD between flowering and fruit harvest maturity, particularly for lower latitude sites, although further validation is warranted. The availability of GDD values collected using a common methodology will facilitate cultivar comparisons (e.g., for selection of cultivars to achieve a desired market window) and in evaluation of the heritability of the trait.

The current study improves on existing GDD recommendations for mango reproductive development for four Australian grown cultivars (Kensington Pride, Calypso, Honey Gold and Keitt), with optimization of Tb and TB values. Recommendations on cultivar specific minimum maturity specifications are given in Table 11. The GDD from Christmas tree to harvest maturity was calculated by subtraction of 180 from the asparagus target for Australian cultivars KP, Honey Gold and Calypso, and 240 for Keitt. The Honey Gold recommendation is approximately 60 units higher, or approx. five calendar days, to the 1500 units recommendation of Winston et al. [4]. The Calypso recommendation is 48 units higher, or approx. four calendar days, to the 1680 units recommendation of Moore et al. [2].

**Table 11.** Recommended minimum harvest maturity specifications by cultivar, based on use of the Upper T calculation of GDD.

| Cultivar | CIE B | Colour Card Equivalent-(Table A3) | DMC (% $w/w$) | SSC: TA | GDD (from Asparagus Stage) | GDD (from Christmas Tree Stage) |
|---|---|---|---|---|---|---|
| KP | 32 (29–34) | 7 | 14.7 | - | 1600 * | 1420 ** |
| Calypso | 32 (29–34) | 7 | 16.0 | 6.5 | 1728 | 1540 |
| Honey Gold | 36 (33–39) | 9 | 18.0 | 6.5 | 1740 | 1560 |
| Keitt | 51 (46–55) | 13 | 16.0 | 6.5 | 2185 | 1936 |

* Diczbalis et al. [1] recommendation; ** Values extrapolated from GDD to reach Christmas tree stage.

The hardware and UpperT method recommended in the current study was implemented in major mango growing areas in Australia, with results viewable on-line given user entered flowering dates (http://fruitronics.com/, accessed on 1 November, 2022). Further work is required to confirm the KP recommendation, and additional work could be carried out for other mango cultivars such as R2E2, NamDocMai and the National Mango Breeding Program cultivars in Australia. Further work could also be carried out to establish a GDD range, involving documentation of loss of storage life with increased harvest GDD.

A single colour card set is recommended for assessment of flesh colour across all cultivars, including Keitt. Cards with scores of 11, 13, and 15 added to the existing 'Calypso' card set (DAF 2019 print), with CIE B values of 43.0, 51.0, and 58.0 (as illustrated in Table A3).

Further study could be carried out to confirm that the delay in maturation of within canopy fruit compared to external canopy fruit is due to a temperature difference. Also, there is variation in flower opening within a panicle, with flower opening typically begins at the base of the panicle and proceeding towards the tip over a period of a week or so, with consequent variation on pollination and fruit set on a given panicle. Future studies could quantify this variation, which adds uncertainty in the GDD forecast of harvest maturity.

There has been some debate within the mango industry on the relative merits of use of flesh colour and DMC in estimation of fruit maturation. As a non-destructive technique, more fruit can be sampled for NIR-DMC than can be destructively assessed for flesh colour. As harvest GDD approaches, it is recommended that growers use the non-destructive measure of NIR-DMC to select fruit of a range of DMC values, and thus maturities, from an orchard. This fruit can then be cut to assess flesh colour as a confirmation of harvest maturity status. The NIR-DMC of fruit at the harvest maturity flesh colour, as evaluated by comparison to colour charts or by use of a chromameter to measure CIE B value, can then be established. That NIR-DMC value can be used in non-destructive assessment of fruit harvest maturity for orchards with similar growing conditions.

**Author Contributions:** Conceptualization, M.H.A. and K.B.W.; methodology, M.H.A. and K.B.W.; investigation, M.H.A.; writing—original draft preparation, M.H.A. and K.B.W.; writing—review and editing, M.H.A., C.M., G.D. and K.B.W.; supervision, K.B.W.; project administration, K.B.W.; funding acquisition, K.B.W. All authors have read and agreed to the published version of the manuscript.

**Funding:** Funding for this project was provided by Hort Innovation from the Australian Government Department of Agriculture, Fisheries and Forestry as part of its Rural R&D for Profit program with Central Queensland University, UNE, Mangoes Australia, NT DITT, NSW DPI and DAF Qld, and by Perfection Fresh, Manbulloo and Pinata. Hort Innovation is the grower-owned, not-for-profit research and development corporation for Australian horticulture. MA acknowledges receipt of a CQU International tuition fee waiver and a living allowance scholarship through Hort Innovation project MG22000.

**Data Availability Statement:** The data presented in this study are available on request from the corresponding author.

**Acknowledgments:** The work formed part of the Master's Thesis of M.A. M.A. acknowledges the contribution in data collection and support given from Maira Aparecida and Paulo Dantas, Agrodan, Brazil, Martina Matzner from Acacia Hills Farms Ltd., Australia. The contribution of Nicholas Anderson in collection of data in 2018 and 2019 and in training for destructive methodologies is acknowledged.

**Conflicts of Interest:** The authors declare no competing interests in the undertaking of this work.

## Appendix A. Panicle Tagging Exercises

**Table A1.** Populations and tagging exercises across different sites and seasons. All panicles were tagged at asparagus stage except as noted.

| Pop (#) | Cultivar | Region | Tagging Date | Panicle Number | Fruit Number | Retention (%) | Harvest Dates | Comments |
|---|---|---|---|---|---|---|---|---|
| **2018** | | | | | | | | |
| 1 | Calypso | Darwin, NT | 29-May | 269 | 95 | 35 | 20/09 | 40 panicles tagged at asparagus, 20 at elongation, 30 at Christmas tree, 5 at fruit set, GDD between asparagus and each stage estimated |
| 2A | Honey Gold | Bugundarra, QLD | 10-Jul | 201 | 31 | 15 | 18/12, 27/12, 31/12 | |
| 2B | Honey Gold | Bugundarra, QLD | 17-Jul | 291 | 73 | 25 | 18/12, 27/12, 31/12 | |
| 2C | Honey Gold | Bugundarra, QLD | 26-Jul | 92 | 47 | 51 | 18/12, 27/12, 31/12 | |
| 2D | Honey Gold | Bugundarra, QLD | 4-Aug | 280 | 30 | 11 | 18/12, 27/12, 31/12 | |
| 2E | Honey Gold | Bugundarra, QLD | 18-Aug | 138 | 28 | 20 | 18/12, 27/12, 31/12 | |
| **2019** | | | | | | | | |
| 3A | Calypso | Darwin, NT | 22-May | 50 | 12 | 24 | 4/10 | |
| 3B | Calypso | Darwin, NT | 29-May | 50 | 24 | 48 | 4/10 | |
| 3C | Calypso | Darwin, NT | 5-Jun | 50 | 15 | 30 | 4/10 | |
| 3D | Calypso | Darwin, NT | 12-Jun | 50 | 11 | 22 | 4/10 | |
| 3E | Calypso | Darwin, NT | 19-Jun | 50 | 9 | 18 | 4/10 | |
| 3F | Calypso | Darwin, NT | 26-Jun | 50 | 21 | 42 | 4/10 | |
| 3G | Calypso | Darwin, NT | 3-Jul | 50 | 1 | 2 | 4/10 | |
| 4A | Honey Gold | Bungundarra, QLD | 16-Jul | 113 | 16 | 14 | 20/12 | |
| 4B | Honey Gold | Bungundarra, QLD | 25-Jul | 60 | 10 | 17 | 20/12 | |
| 4C | Honey Gold | Bungundarra, QLD | 2-Aug | 18 | 7 | 39 | 20/12 | |
| 4D | Honey Gold | Bungundarra, QLD | 14-Aug | 30 | 13 | 43 | 20/12 | |
| 4E | Honey Gold | Bungundarra, QLD | 30-Aug | 24 | 2 | 8 | 20/12 | |

**Table A1.** *Cont.*

| Pop (#) | Cultivar | Region | Tagging Date | Panicle Number | Fruit Number | Retention (%) | Harvest Dates | Comments |
|---|---|---|---|---|---|---|---|---|
| **2020** | | | | | | | | |
| **5A** | Honey Gold | Bungundarra, QLD | 5-Jul | 600 | 55 | 9 | 03/12, 10/12, 17/12, 23/12, 28/12/2020, 07/01, 15/01/2021 | |
| **5B** | Honey Gold | Bungundarra, QLD | 5-Aug | 600 | 57 | 10 | 03/12, 10/12, 17/12, 23/12, 28/12/2020, 07/01, 15/01/2021 | |
| **6A** | Honey Gold | Bungundarra, QLD | 5-Jul | 600 | 58 | 10 | 03/12, 10/12, 17/12, 23/12, 28/12/2020, 07/01, 15/01/2021 | |
| **6B** | Honey Gold | Bungundarra, QLD | 13-Aug | 600 | 47 | 8 | 03/12, 10/12, 17/12, 23/12, 28/12/2020, 07/01, 15/01/2021 | |
| **7** | Keitt | Bungundarra, QLD | 5-Aug | 300 | 71 | 24 | 22/12, 29/12/2020, 07/01, 13/01, 20/01, 29/01, 04/02/2021 | |
| **8** | Keitt | Bungundarra, QLD | 25-Aug | 200 | 49 | 25 | 20/01, 29/01, 04/02, 11/02, 10/03/2021 | |
| **9** | Keitt | Belem do Sao Francisco, Brazil | 6-Aug | 300 | 81 | 27 | 28/12, 31/12/2020, 05/01/2021 | Tagged by farm staff, first samples were cool stored for a week before assessing |
| **2021** | | | | | | | | |
| **10** | Calypso | Darwin, NT | 4-Jun | 100 | 44 | 44 | 28/09, 01/10, 04/10 | |
| **11** | Calypso | Darwin, NT | 4-Jun | 100 | 27 | 27 | 23/09 | tagged at Christmas tree stage |
| **12** | Honey Gold | Darwin, NT | 8-Jul | 50 | 13 | 26 | 27/10 | |
| **13** | Honey Gold | Katherine | 2-Jul | 100 | 40 | 40 | 26/10 | |
| **14** | Calypso | Katherine, NT | 15-Jun | 100 | 9 | 9 | 20/10 | |
| **15** | KP | Katherine, NT | 15-Jun | 50 | 13 | 26 | 7/10 | |
| **16** | Calypso | Dimbulah, QLD | 30-Jun | 100 | 29 | 29 | 25/11, 29/11 | 5 fruit were not destroyed at harvest and used in a ripening exercise |
| **17** | Honey Gold | Dimbulah, QLD | 30-Jun | 100 | 20 | 20 | 29/11 | 5 fruit were not destroyed at harvest and used in a ripening exercise |

**Table A1.** *Cont.*

| Pop (#) | Cultivar | Region | Tagging Date | Panicle Number | Fruit Number | Retention (%) | Harvest Dates | Comments |
|---|---|---|---|---|---|---|---|---|
| **18** | Keitt | Belem do Sao Francisco, PE, Brazil | 16-Jun | 100 | 32 | 32 | 8/11 | 17 fruit do not have CIE LAB readings |
| **19** | Keitt | Curaca, BA. Brazil | 18-Jun | 100 | 24 | 24 | 15/11 | 9 fruit do not have CIE LAB readings |
| **20** | KP | Bungundarra, QLD | 5-Jul | 50 | 2 | 4 | 1/12 | all internal fruit (external fruit loss from chilling injury) |
| **21** | Honey Gold | Bungundarra, QLD | 24-Jun | 100 | 5 | 5 | 8/12 | all internal fruit (external fruit loss from chilling injury) |
| **22** | Keitt | Bungundarra, QLD | 5-Jul | 100 | 5 | 5 | 5/01 | suffered loss from bacterial black spot |

## Appendix B. Flesh Colour as Maturity Targets

**Table A2.** CIE B value of mango harvest maturity colour cards. Data is presented for a Calypso card set of the original (2010) and of a second printing (2019), and values from the pdf of the file sent to the printer for the original printing. However, the cards of the second printing had been heavily used in field and were 'aged'. Data for Keitt is from the pdf file of card swatches and from print. Data is presented of three original Kensington Pride 'business cards'. Mean and SE of three replicate readings is presented.

| DAF Calypso Picking Guide | Card Set 1 (2019 Version)– Second Printing | Card Set 2 (2010 Version)–Original Printing | Card Set 3 (2010 Version)-Digital |
|---|---|---|---|
| 3 | 19.1 ± 0.2 | 17.5 ± 0.3 | 17.0 ± 0.0 |
| 5 | 25.7 ± 0.1 | 22.5 ± 0.3 | 26.0 ± 0.0 |
| 7 * | 32.4 ± 0.1 | 26.2 ± 0.5 | 34.0 ± 0.0 |
| 9 | 35.8 ± 0.1 | 34.6 ± 0.5 | 43.0 ± 0.0 |
| 11 | 41.6 ± 0.1 | 43.1 ± 0.6 | not available |
| KP 'business' card | Card set 1 | Card set 2 | Card set 3 |
| Mature mango * | 31.0 ± 0.0 | 30.9 ± 0.0 | 31.0 ± 0.0 |
| Keitt, US Mango Board Maturity and Ripeness guide | Digital version | Printed 1 | Printed 2 |
| 1 | 43.0 ± 0.1 | 34.4 ± 0.2 | 34.3 ± 0.2 |
| 2 * | 51.0 ± 0.1 | 45.8 ± 0.2 | 45.8 ± 0.2 |
| 3 | 58.0 ± 0.1 | 48.5 ± 0.2 | 48.6 ± 0.2 |
| 4 | 66.0 ± 0.1 | 50.5 ± 0.2 | 50.5 ± 0.2 |
| 5 | 75.0 ± 0.1 | 53.0 ± 0.2 | 52.9 ± 0.2 |

* The colour recommended as denoting maturity is denoted by an asterisk.

**Table A3.** Proposed fruit maturity colour swatches and associated CIE LAB values, with maturity targets specific to cultivar. CIE L have two proposed values separated by slash (/), the first for printing purposes based on use of a PDF document, the second being the expected CIE L reading of fruit flesh.

| 3 CIE L = 97.00/85.00 CIE A = −4.40 CIE B = 18.00 | 5 CIE L = 97.00/85.00 CIE A = −5.20 CIE B = 26.00 | 7 (Calypso/KP) CIE L = 97.00/85.00 CIE A = −5.70 CIE B = 32.00 | 9 (Honey Gold) CIE L = 95.00/84.00 CIE A = −5.90 CIE B = 36.00 |
|---|---|---|---|
| | | | |
| 11 CIE L = 95.00/84.00 CIE A = −4.70 CIE B = 43.00 | 13 (Keitt) CIE L = 92.00/83.00 CIE A = −3.80 CIE B = 51.00 | 15 CIE L = 92.00/83.00 CIE A = −1.08 CIE B = 57.00 | 17 CIE L = 88.00/80.00 CIE A = 1.60 CIE B = 62.00 |
| | | | |

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
