# Peer review of "Growing Degree Day Targets for Fruit Development of Australian Mango Cultivars"

_horticulturae, doi:10.3390/horticulturae9040489_

Round 1

Author Response

The authors improved the cumulative growth degree days (GDD) target from early flowering

to fruit harvest maturity for the prediction of mango harvest time. In addition, a color card for maturity assessment was proposed. The data has been acquired and validated at multiple

locations in four varieties. This is a practical study.

>>Agreed, the paper has practical value.

However, there is little discussion of the characteristics of the target varieties, making it difficult for the reader to determine whether the research can be applied to other regions and multiple varieties, or not. I would appreciate universal consideration of what you have identified in this paper.

>> GDD targets are cultivar specific but should be location independent. There is ‘universality’ in the recommendation of the GDD algorithm using an upper base temperature, in the recommendation of colour space values of a set of colour cards for use across all cultivars and in the recommendation of a method to follow to estimate GDD for any cultivar. The latter recommendation has already been taken up, with the method followed by one the Australian state Dept of Agriculture using the protocol for GDD estimates of other cultivars.  Explanatory text has been added to the manuscript to highlight these points.

Below are comments on individual sentences,

Abstract

Please insert the scientific name.  >>done

L15-16  Tb、TB Please define as in line 150-151. >>done

L163 Please explain why these four varieties were selected. Are there differences with respect to genetic background, maturation period and color etc.? >>description has been added (at li 167)

L343-345 You say that “The difference between the GDD requirement of the  three Australian cultivars, which share parentage, and the Florida cultivar, Keitt, is consistent with a genetic component to the GDD requirement. Describe that information in the Introduction. >>Description has been added in the paragraph beginning at li 167

L179 Is there a model number? Usually, for accurate outdoor temperature measurements, ventilator (a device combining an awning and a ventilation fan) is used to avoid solar radiation and other radiative effects. Please explain a little more about the structure of the sensor. >>Description has been added (at line 197)

L383 Figure 4 Text and plots in the figure are hard to see. Please devise a way to make the colors and shapes of the plots easily identifiable. Does the orientation of the red line have any significance?

>>Figures have been re-done (and clarity improved) in SigmaPlot. The red line is now clearly explained in the figure legend.

L374-460

There is a lengthy description of the results. Please restructure paragraphs.

>> We have attempted some revision of the Results, but, as the reviewer have noted, this manuscript reports on a large body of work. We have not been successful in shortening the description of the results.

Reviewer 2 Report

The paper is very extensive, but this is due to the number of results presented, which favours an assessment of both the methodology used and the amount of work required to carefully conduct the study.

In particular, the frankness of the description of the discrepancies in the timing of the panicles tagging in the different experimental locations (lines 197-208) should be appreciated, as many authors prefer to present only a uniform general scheme of experiments.

Major comments

Readers who do not regularly use the different versions of GDD will be frustrated by the lack of explanation of the relevance of TB to the idea of GDD. Such information is only found in lines 330-332, but should be in the introduction already, together with a reminder of the importance of Tb.

Minor comments

Lines 34 and 35: [2] and [3] are not sufficient when citing literature

Line 61: Does NT stand for Northern Territory? I'm not Australian and I'm not sure.

Line 96: The word “ratio” is missing after “SSC: TA”

Table 2: What is the “KP” in the “Cultivar” column?

Lines 129-130: What is the “Australian National Mango Plant Breeding Program cultivar”?

Lines 157-158: Unfinished description “weather screen placed 1.2 m”

Lines 247-248: Several words repeated in one sentence

Lines 249-250: Arcus tangent function should be notated as “arctan” rather than “tan-1”, as notation “tan-1” is often understood as 1/tan

Line 337: Table number is missing

Figure 4 is not legible. Labels on the graphs’ axes and scale values are often too small. The colours are not varied enough and all the markers are the same. If the authors do not want to change some of the colours, at least the line styles should be different.

Lines 395 and 396: There is no data for the Kensington Pride cultivar in Figure 4.

Author Response

The paper is very extensive, but this is due to the number of results presented, which favours an assessment of both the methodology used and the amount of work required to carefully conduct the study.

>>Thank you

In particular, the frankness of the description of the discrepancies in the timing of the panicles tagging in the different experimental locations (lines 197-208) should be appreciated, as many authors prefer to present only a uniform general scheme of experiments.

>>Thank you for this recognition. This is the reality of working across 22 commercial sites spread over thousands of kilometers.

Major comments

Readers who do not regularly use the different versions of GDD will be frustrated by the lack of explanation of the relevance of TB to the idea of GDD. Such information is only found in lines 330-332, but should be in the introduction already, together with a reminder of the importance of Tb.

>> Explanatory text has been added to the Introduction, as suggested

Minor comments

Lines 34 and 35: [2] and [3] are not sufficient when citing literature

>>A full review of literature on mango GDD is provided in Table 1 and preceeding text. The sentence at (now line 58) is provided as an example (“For example, for cultivar Calypso, Moore [2] recommended 1680 GDD …….) of confusion in practical application when starting flowering stage and Tb values are varied. We believe the rewoding undertaken clarifies this point.

Line 61: Does NT stand for Northern Territory? I'm not Australian and I'm not sure.

>>This abbreviation is now spelt out.

Line 96: The word “ratio” is missing after “SSC: TA”

>>done

Table 2: What is the “KP” in the “Cultivar” column?

>> now explained in legend

Lines 129-130: What is the “Australian National Mango Plant Breeding Program cultivar”?

>>sentence reworded to improve clarity

Lines 157-158: Unfinished description “weather screen placed 1.2 m”

>>sentence reworded to improve clarity

Lines 247-248: Several words repeated in one sentence

>> Our mistake! corrected

Lines 249-250: Arcus tangent function should be notated as “arctan” rather than “tan-1”, as notation “tan-1” is often understood as 1/tan

>>corrected

Line 337: Table number is missing

>>corrected (now line 391)

Figure 4 is not legible. Labels on the graphs’ axes and scale values are often too small. The colours are not varied enough and all the markers are the same. If the authors do not want to change some of the colours, at least the line styles should be different.

>> figure has been redrawn (in Sigmaplot) to improve clarity

Lines 395 and 396: There is no data for the Kensington Pride cultivar in Figure 4.

>> Figure 4 is introduced at line 430 as providing an example of three data sets, to explain the method followed. After this, a li 468, we explain “This procedure was followed with all calibration data sets, to associate attribute levels to the available harvest maturity targets of flesh colour in Calypso, KP….”.  We believe, that in context of the general wording improvements made, this explanation is clear to the reader.

Reviewer 3 Report

The manuscript titled "Growing Degree Day Targets for Fruit Development of Australian Mango Cultivars" focuses on improving existing GDD recommendations for mango reproductive development for some Australian grown cultivars, with comparison of a GDD calculation employing a minimum base temperature only and a calculation using both a minimum (Tb) and maximum (TB) base temperature, with optimization of Tb and TB values. The paper is well written and clear. It is interesting for readers due to the fact that it deals with an innovative theme related to the management of the farm based on the timing of the mango harvest.  It needs some minor revisions before being considered for publication. The introduction needs to be improved. The authors should better present the topic studied by focusing on the species and on the choice of cultivars. The problem of estimating harvest times to plan and improve the marketing farm should be further considered in this section. Furthermore, the statistical analysis paragraph should also be more detailed, specifying which tests were used. It is suggested to make the conclusions more concise in order to explain the study hypotheses made more clearly. Finally, the bibliography must be formatted following the guidelines of the journal. As for the choice of the special issue, I have some doubts. I think there may be others that fit better, but I leave the decision to the editor.

Best regards

Author Response

The manuscript titled "Growing Degree Day Targets for Fruit Development of Australian Mango Cultivars" focuses on improving existing GDD recommendations for mango reproductive development for some Australian grown cultivars, with comparison of a GDD calculation employing a minimum base temperature only and a calculation using both a minimum (Tb) and maximum (TB) base temperature, with optimization of Tb and TB values. The paper is well written and clear. It is interesting for readers due to the fact that it deals with an innovative theme related to the management of the farm based on the timing of the mango harvest. 

>> thank you

It needs some minor revisions before being considered for publication. The introduction needs to be improved. The authors should better present the topic studied by focusing on the species and on the choice of cultivars. The problem of estimating harvest times to plan and improve the marketing farm should be further considered in this section.

>> The Introduction has been revised, with particular focus on the cultivars employed.  abstract notes “The current study was undertaken to improve GDD targets for Australian mango cultivars”

Furthermore, the statistical analysis paragraph should also be more detailed, specifying which tests were used. It is suggested to make the conclusions more concise in order to explain the study hypotheses made more clearly. Finally, the bibliography must be formatted following the guidelines of the journal. As for the choice of the special issue, I have some doubts. I think there may be others that fit better, but I leave the decision to the editor.

>>Section 2.6 (Statistical analysis) has been rewritten.  We have attempted some revision of the Conclusion in the spirit of the reviewer suggestion, but we have not been successful in shortening this section. References are in MDPI format (MDPI EndNote style adopted).

Reviewer 4 Report

A very interesting multi-year experience, with different locations in two countries. The authors are asked to standardize the notation of data both in the text of the publication and in the tables. Lack of precise specification of the experiment layout, number of repetitions and plants per plot. How many samples were taken, from how many plants, from what places on the plant. How many trials in laboratory tests. Repeating the same verbs very often, even in one task. No explanation of some abbreviations (Tb, TB, TN, v/w, w/w). Valuable results, after eliminating the marked errors or explaining them, can be successfully published

Author Response

A very interesting multi-year experience, with different locations in two countries.

>> thank you

The authors are asked to standardize the notation of data both in the text of the publication and in the tables. Lack of precise specification of the experiment layout, number of repetitions and plants per plot. How many samples were taken, from how many plants, from what places on the plant. How many trials in laboratory tests. Repeating the same verbs very often, even in one task. No explanation of some abbreviations (Tb, TB, TN, v/w, w/w).

>> The text has been edited to address these issues. Specifically, detail of all populations is included in an Appendix, while number of panicles tagged is now clearly described in the manuscript text. Abbreviations are described at first use.

Valuable results, after eliminating the marked errors or explaining them, can be successfully published

>> Thank you

Please find attached tracked changes If interested
